# Structure of the gene therapy vector, adeno-associated virus with its cell receptor, AAVR

Nancy L Meyer[1†], Guiqing Hu[2†], Omar Davulcu[1], Qing Xie[1], Alex J Noble[2‡], Craig Yoshioka[3], Drew S Gingerich[3], Andrew Trzynka[1], Larry David[1], Scott M Stagg[2,4], Michael Stewart Chapman[1,5]*

[1]Department of Biochemistry and Molecular Biology, Oregon Health and Science University, Portland, United States; [2]Institute Molecular Biophysics, Florida State University, Tallahassee, United States; [3]OHSU Center for Spatial Systems Biomedicine, Portland, United States; [4]Department of Chemistry and Biochemistry, Florida State University, Tallahassee, United States; [5]Department of Biochemistry, University of Missouri, Columbia, United States

**Abstract** Adeno-associated virus (AAV) vectors are preeminent in emerging clinical gene therapies. Generalizing beyond the most tractable genetic diseases will require modulation of cell specificity and immune neutralization. Interactions of AAV with its cellular receptor, AAVR, are key to understanding cell-entry and trafficking with the rigor needed to engineer tissue-specific vectors. *Cryo*-electron tomography shows ordered binding of part of the flexible receptor to the viral surface, with distal domains in multiple conformations. Regions of the virus and receptor in close physical proximity can be identified by cross-linking/mass spectrometry. *Cryo*-electron microscopy with a two-domain receptor fragment reveals the interactions at 2.4 Å resolution. AAVR binds between AAV's spikes on a plateau that is conserved, except in one clade whose structure is AAVR-incompatible. AAVR's footprint overlaps the epitopes of several neutralizing antibodies, prompting a re-evaluation of neutralization mechanisms. The structure provides a roadmap for experimental probing and manipulation of viral-receptor interactions.
DOI: https://doi.org/10.7554/eLife.44707.001

*For correspondence:
chapmanms@missouri.edu

†These authors contributed equally to this work

Present address: ‡National Resource for Automated Molecular Microscopy, Simons Electron Microscopy Center, New York Structural Biology Center, New York, United States

**Competing interests:** The authors declare that no competing interests exist.

## Introduction

The human parvovirus, AAV, has a 60-subunit protein capsid shell containing a single-stranded DNA genome (*Xie et al., 2002*; *Chapman and Agbandje-Mckenna, 2006*). Recombinant AAV is used as a cellular delivery vehicle in emerging clinical applications of gene therapy: Luxturna is the first in vivo gene therapy approved for clinical treatment in the US, with demonstrated efficacy combatting a retinal dystrophy that leads to loss of vision (*Russell et al., 2017*). AAV's interactions on cell entry are of fundamental virological interest and provide a foundation for engineering more efficient and cell-specific delivery vectors needed to treat an array of diseases (*Kotterman and Schaffer, 2014*).

Infection starts with AAV's attachment to serotype-specific glycan 'primary' receptors, followed by co-receptor-mediated endocytotic entry. AAV2, the type species studied here, like several other AAVs, attaches to heparan sulfate proteoglycan (HSPG) (*Summerford and Samulski, 1998*; *Kern et al., 2003*; *Opie et al., 2003*; *Huang et al., 2014*). Recently, through genome-wide screening for genes essential to transduction, the cellular protein AAVR was implicated as the key receptor for entry of a panel of AAV serotypes into representative cell types and, in vivo, in mice (*Pillay et al., 2016*). AAVR is a membrane protein whose glycosylation is not essential for binding or cell-transduction by AAV (*Pillay et al., 2016*; *Pillay et al., 2017*). From N- to C-terminus, it comprises (*Figure 1*)

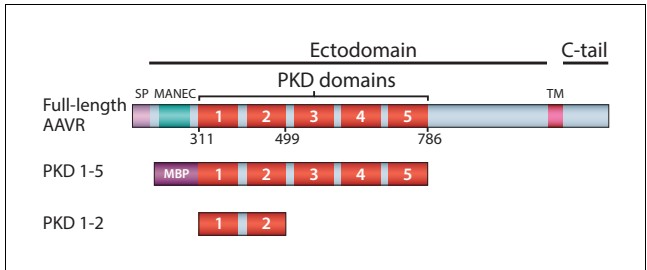

**Figure 1.** Schematic representation of the domain structure of AAVR, and the AAV-binding constructs used in this work. Domain acronyms: SP - signal peptide; MANEC - motif at the N-terminus with eight cysteines; PKD - polycystic kidney disease; TM – transmembrane helix; C-tail – cytoplasmic, C-terminal tail; MBP – maltose-binding protein (fusion). Numbers refer to the first or last amino acids of native AAVR used in the construct.
DOI: https://doi.org/10.7554/eLife.44707.002

a signal peptide, a MANEC domain, five immunoglobulin-like PKD domains (PKD1-5) (*Ibraghimov-Beskrovnaya et al., 2000*), a transmembrane region and a small cytoplasmic domain. Over-expression of 'mini-AAVR' (PKD domains 1–3, but lacking MANEC) can support transduction in an AAVR knock-out, while AAV-binding and transduction inhibition are both achieved with a fusion of maltose binding protein and the five PKD domains (MBP-PKD1-5) (*Pillay et al., 2016*). Thus, structural studies of an AAV complex were begun with this soluble construct. Intriguingly, in concurrent studies, it emerged that PKD domain 2 (PKD2) was most critical for AAV2's interactions, in contrast to PKD1 for AAV5 (*Pillay et al., 2017*). Here, *cryo*-ET, together with single particle *cryo*-EM and cross-linking analysis, reveal that AAVR binds tightly to the AAV2 viral surface through well-defined interactions with PKD2; PKD1 is more loosely associated, and the membrane-proximal domains (PKD3-5), distal from the virus surface, have varied/flexible conformation.

## Results

### Strategic plan

A central objective was to visualize, at the highest resolutions possible, the structure of AAV complexed with the cellular receptor, AAVR, which is essential for entry of most AAVs. AAVR is an integral membrane protein whose interactions with AAV are mediated through its ectodomain regions (*Pillay et al., 2016*). AAVR would be expressed heterologously for preparation of soluble complexes. Even omitting the transmembrane and cytoplasmic C-terminal regions, constructs would be multi-domain, a potential source of conformational heterogeneity. It was also not known whether receptor-binding would conform to the icosahedral symmetry of the virus, particularly because the capsid subunits are not identical. Most capsid proteins are viral protein (VP) 3, but alternative splicing leads to about 10% fractions of two N-terminally extended variants, VP1 and VP2 (*Berns, 1996*). The unique regions of VP1 and VP2 have not been resolved in crystal structures (*Xie et al., 2002*). If key to AAVR interactions, it was not known to which subset of the otherwise indistinguishable 60 subunits AAVR would be bound, and whether they would be in the same locations on every virion. Such a large potentially heterogeneous complex was not a good candidate for crystallographic analysis, and, could pose challenges for electron microscopy (EM).

The prospects for EM at near atomic resolution throughout the entire receptor seemed low, so diverse approaches were planned for hybrid-methods structure determination. Biomolecular EM is limited by the electron dose usable for imaging before excessive radiation damage of the sample, even when mitigated at cryogenic temperatures (*cryo*-EM) (*Chen et al., 2008*). The primary data in EM are 2D projection images of 3D samples, from which 3D maps are reconstructed by one of two approaches (*Thompson et al., 2016*): In electron tomography (ET), a series of images is collected as the sample is tilted, so that a 3D map can be computed as in CT (medical) scanning. 3D information is obtained for each particle on the sample grid, but spreading the acceptable low dose of electrons over 100 + images limits the resolution achievable. In more common single-particle analysis (SPA), each of many particles is imaged just once with all the available electron dose. 3D maps, of

potentially higher resolution, are reconstructed from many particles providing that they share sufficiently identical conformation and that the orientation of each can be determined from a single 2D image.

In strategizing how best to image AAV2-AAVR complexes, there were multiple considerations. If and where the receptor conformed to the symmetry of AAV, high-resolution *cryo*-EM might be possible, through SPA, enhanced by application of icosahedral symmetry. Should there be heterogeneity in the location within the capsid of receptor-bound AAV2 subunits, or in the conformation of any receptor domains that are not interacting directly with the virus, then *cryo*-ET might be more appropriate. Sub-tomogram reconstruction would facilitate classification (in 3D) of the many potential configurations, before signal averaging. Resolutions possible with *cryo*-ET are lower than *cryo*-EM, and generally insufficient to recognize individual domains. Atomic modeling is usually only possible for *cryo*-EM, and sometimes only possible for *cryo*-ET with additional constraints coming from other biophysical characterizations. A hybrid approach was planned, identifying regions of the virus and receptor primary structures that are in close proximity, through chemical cross-linking, proteolysis and tandem mass spectrometry. The known crystal structure of AAV2 could be overlaid on the EM reconstruction by alignment of point group symmetry. A homology model of the receptor could then be docked approximately, by fitting the low-resolution *cryo*-ET density while satisfying distance constraints from the cross-linking. Between the resolution regimes of conventional *cryo*-EM and *cryo*-ET lay the emerging approach of SPA combined with computational sub-volume extraction (*Ilca et al., 2015*). All three approaches were pursued in parallel, because, at the start, the extent of heterogeneity in virus-receptor complexes and the imaging resolutions achievable were uncharacterized.

Heterogeneity from inter-domain receptor flexibility might also be addressable through biochemical elimination of domains not essential to AAV-binding. Our emerging cross-linking results, together with biochemical and genetic characterizations (*Pillay et al., 2017*), indicated which AAVR domains were most important. This allowed a divide-and-conquer strategy, high resolution coming from minimal constructs, with context and relevance to the native receptor coming from lower resolution studies with larger receptor constructs. Thus, redundant tracks were pursued, testing different combinations of domains to find the complexes most suitable for high resolution structure, then validating by comparison to lower resolution reconstructions with near-native receptor constructs.

## *Cryo*-electron tomography of AAV2 complexed with a PKD1-5 fusion protein

Structural studies started with the MBP-PKD1-5 construct, known to contain the AAV-binding elements (*Pillay et al., 2016*). However, single-particle reconstruction of the AAV complex showed only virus density: AAVR density would be weakened on application of the viral 60-fold symmetry, if only a fraction of binding sites were occupied. Up to three bound AAVRs were seen in raw tomograms (*Figure 2A*), but their unpredictable distribution over the viral surface confounded automatic interparticle alignment: AAVR was again washed out in the whole-virus sub-tomogram average at ~10 Å resolution. AAVR density was seen in a reconstruction made by manually marking their locations in tomograms, then averaging only occupied sites on the virus using known icosahedral symmetry operators. From eight tilt series, 2602 of the 60 sites in 1321 particles (3.3%) were occupied, averaging 1.97 receptors/virion. Multivariate data analysis (MDA) and ascendant classification yielded 20 classes, four commensurate with AAVR's size. Modest numbers of particles per class (~150) limited resolution to ~30 Å. The four classes indicated a single binding interface near spikes surrounding the threefold axes, with more varied conformation of distal AAVR domains that were less constrained by viral interactions (*Figure 2B,C*; *Figure 3*).

## Single-particle *cryo*-EM and subvolume extraction for AAV2 complexed with a PKD1-5 fusion protein

Higher resolution was attempted, switching from tomography to single particle analysis (SPA), with classification of smaller sub-volumes local to each threefold axis (*Ilca et al., 2015*). Receptor-bound sites could be distinguished from unoccupied, with a single domain of AAVR bound tightly to AAV, at the location seen by tomography, but now at ~10 Å resolution (*Figure 2E*). Discrete

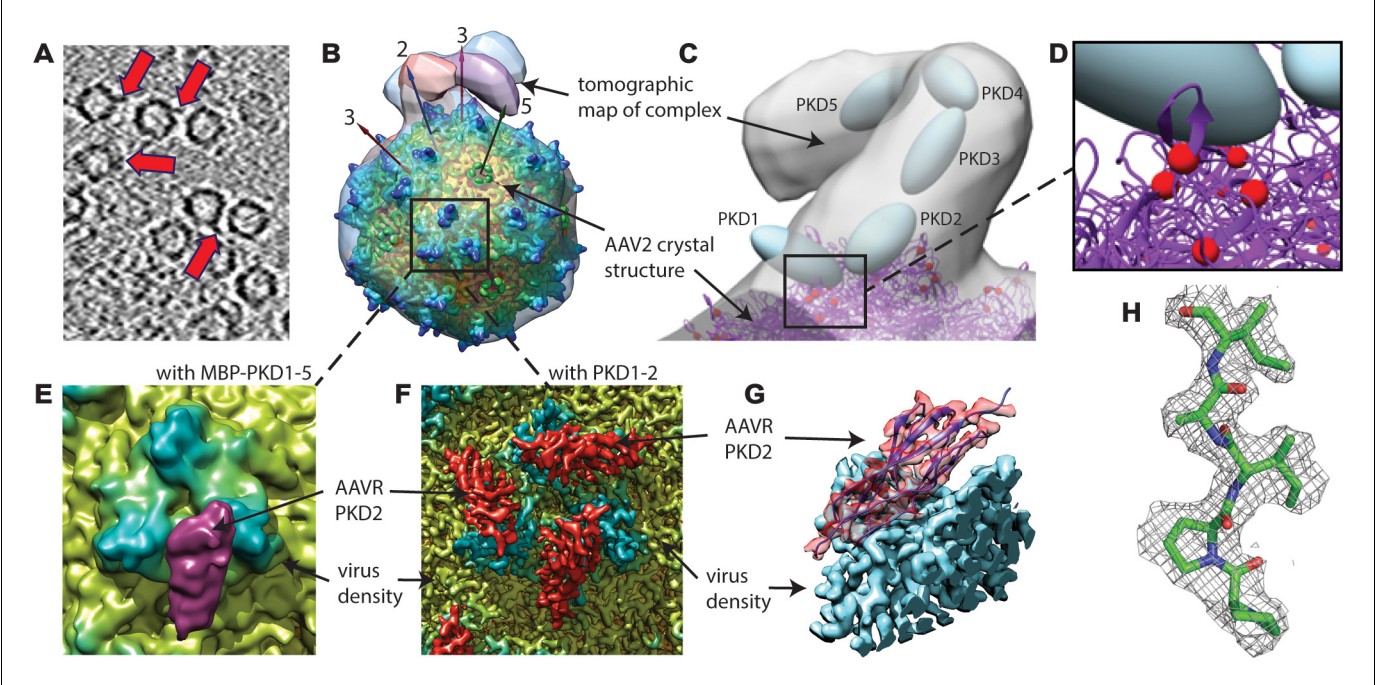

**Figure 2.** EM imaging of AAV-AAVR complexes. (**A**) A slice of a raw tomogram of a complex between AAV2 and the MBP-PKD1-5 fusion construct highlighted with arrows. (**B**) Four of the sub-tomogram classes (*Figure 3*) that result from aligning occupied sites, overlaid on the crystal structure of AAV2 (*Xie et al., 2002*) with selected symmetry axes numbered. AAVR binds near the threefold spikes with varied configuration of viral-distal domains. (**C**) Domain-sized ellipsoids are modeled into the highest-population tomographic class at ~30 Å resolution. (**D**) AAV2 sites (red) that can be cross-linked to PKD1 or the PKD1/2 hinge. (**E**) The sub-volume classified single particle reconstruction of the MBP-PKD1-5 fusion complex shows one domain (purple), ordered and bound on the shoulder and plateau between spikes surrounding each threefold. (**F**) The icosahedrally-averaged single particle reconstruction of a complex with PKD1-2 shows binding at a larger fraction of the same site and its symmetry-equivalents. (**G**) Rotated to a tangential view, the PKD density (red) is traceable as an immunoglobulin domain; (**H**) Resolution of 2.4 Å is sufficient to model side chains specific to the PKD2 sequence.

DOI: https://doi.org/10.7554/eLife.44707.003

conformations of other domains were not resolved by classification, so shorter AAVR constructs were used henceforth to reduce heterogeneity.

## Single-particle *cryo*-EM for AAV2 complexed with a PKD1-2 construct

Deletion mutants and binding studies (*Dudek et al., 2018*) were implicating PKD domains 1–2 as most critical. Size exclusion chromatography revealed gradual oligomerization of several AAVR constructs (*Figure 4*), so AAV was adhered to carbon-coated EM grids (*O'Donnell et al., 2009*), then freshly fractionated His$_6$-tagged PKD1-2 was added. The smaller PKD1-2 construct yielded AAV2 complexes that were found empirically (see below) to have more saturated binding at symmetry-equivalent sites. One can speculate that the absence of PKD3-5, and of the MBP fusion domain, reduced the potential for steric conflict between AAVR molecules bound to adjacent sites on the viral surface. It is also possible, with the propensity of AAVR to oligomerize (*Figure 4*), and perhaps to thereby cross-link AAV2 particles, that the picking of well-separated particles from images of complexes formed in free solution, might have biased our earlier reconstructions towards low-occupancy particles. Aggregation and biased sampling would be minimized with the new approach of adhering AAV2 to the EM grid before adding AAVR, and this perhaps also contributed to the higher binding saturation that was obtained.

The more highly saturated binding allowed single particle processing with icosahedral symmetry, yielding a reconstruction with an FSC$_{0.143}$ of 2.4 Å (*Figure 5*). A single domain was revealed, fully consistent with the tomography and SPA of the PKD1-5 complex, but now resolving the backbone trace and most side chains (*Figure 2F–H*), allowing assignment of the tightly-bound domain as PKD2. Conservative atomic refinement, constraining receptor and virus B-factors to be equal, gives

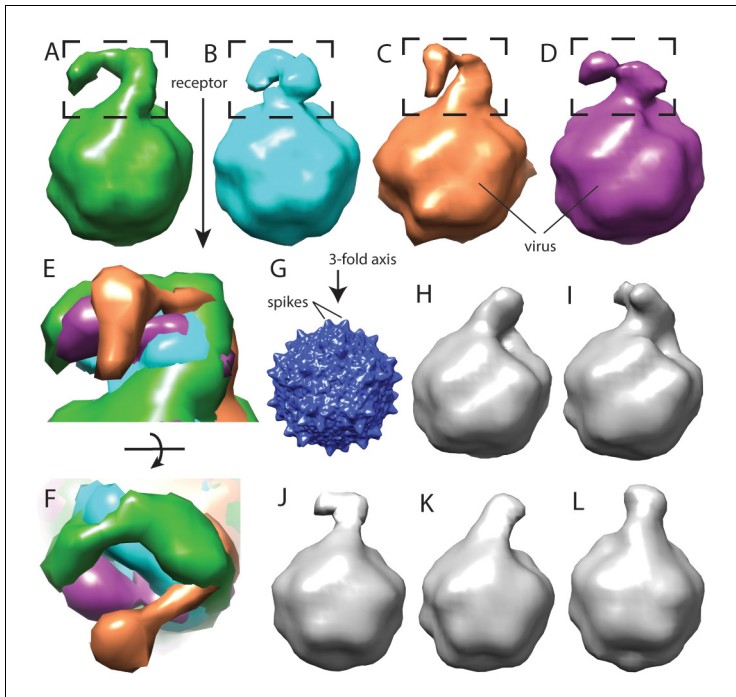

**Figure 3.** Flexibility between domains of AAVR, revealed by cryo-electron tomography. (**A–D**) Class averages, oriented like the overall symmetry-averaged reconstruction (**G**), correspond to classes 1 through 4 of EMDB depositions EMD-0621 through EMD-0624, respectively. They show AAVR anchored at the same location on AAV, near three spikes surrounding symmetry axes on the AAV surface, but with distal PKD domains in different orientations; (**E** and **F**) Tangential and top-down magnifications of the boxed region of classes (**A – D**) superimposed, highlighting variation in domain conformations; (**H–L**) Additional classes for which viral-distal PKD domains are unseen, presumably due to disorder from inter-domain flexibility.

DOI: https://doi.org/10.7554/eLife.44707.004

a lower bound PKD2 occupancy of 0.48. When B-factors are refined, they account for the expected higher disorder in the receptor ligand ($\langle B_{AAV2}\rangle$=15.2 $\text{Å}^2$; $\langle B_{AAVR}\rangle$=27.7 $\text{Å}^2$), and occupancy refines to 0.59. If one speculates that AAV2 particles, adhered to an EM grid, rest on three spikes, then binding of PKD1-2 would be occluded at minimally 9 of 60 (15%) symmetry-equivalent sites. The occupancy of 0.59 therefore corresponds to binding at 70% of remaining available sites. The empirical stoichiometry of approximately two AAVR constructs per three AAV2 subunits provides evidence that the binding of the shortened PKD1-2 construct at adjacent sites on the virus is not completely excluded. At the end of atomic refinement, the model map correlation coefficient is 0.88 after optimization of a low-pass filter applied to the atomic density, giving a $d_{1/2}$ = 2.3 Å that is close to the $FSC_{0.143}$ = 2.39 Å (*Figure 5*) and is indicative of the effective resolution (*Chapman et al., 2013*).

## Cross-linking studies

Cross-linking, with sites identified through mass spectrometry (MS), validated the PKD2 atomic model and improved the approximate placement of PKD1 relative to the tomographic density. The 13.4 Å distance between AAVR-Lys$_{93}$ N$_\zeta$ and AAV2-Lys$_{556}$ in the PKD1-2 complex (that was not cross-linked in the EM sample) is consistent with an MS experiment in which AAVDJ was cross-linked using CBDPS (14 Å spacer) to an AAVR construct missing only the transmembrane and C-terminal tail (*Table 1*). A homology model of PKD1, anchored by the high-resolution PKD2 structure, can be adjusted to satisfy simultaneously 3 of 4 CBDPS cross-links between AAV2 and MBP-PKD1-5, and to explain diffuse (disordered) EM density above each 2-fold axis that is suggestive of multiple/asymmetric configurations (*Table 1*; *Figure 2C,D*). Extending C-terminally from the high-resolution PKD2 structure, PKD domains 3–5 could be fit into the low-resolution sub-tomograms, projecting away from AAV in a variety of conformations (*Figure 2C*, *Figure 3*).

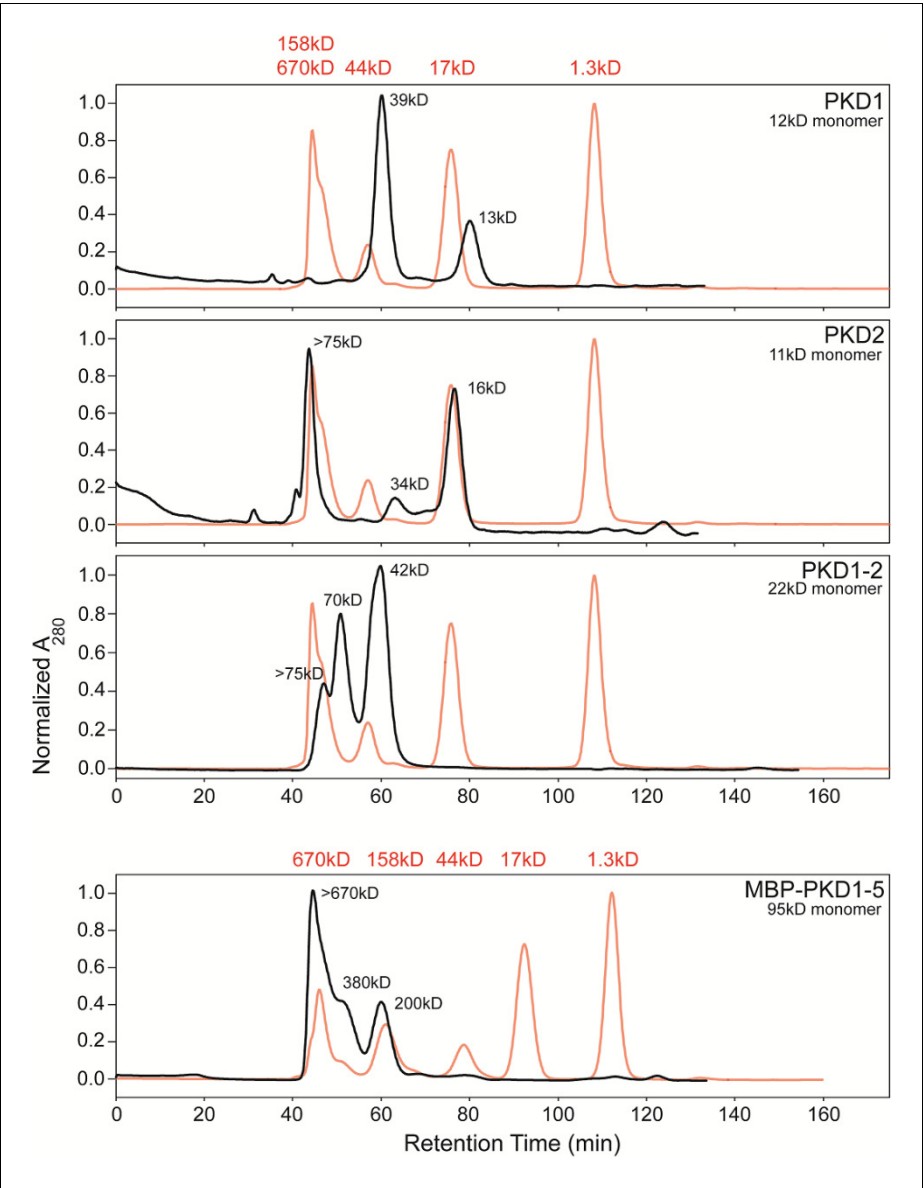

**Figure 4.** Size exclusion chromatography allowing fractionation of oligomeric states of His$_6$-tagged AAVR-PKD constructs. PKD1, PKD2, and PKD1-2 were run on a Superdex 75 column and MBP-PKD1-5 on Superdex 200, and are compared to standards, in red. MBP-PKD1-5 revealed hexamers predominating over tetramers and dimers, which are also seen in PKD1-2 along with larger oligomers. Monomers were only seen with PKD1 and PKD2 (along with larger species). Separated fractions of PKD1-2 re-equilibrated over a month at 4°C, inspiring EM sample preparation by first adhering AAV to a thin carbon film on the EM grid (*O'Donnell et al., 2009*) prior to addition of a freshly-prepared chromatographic fraction of an AAVR construct.
DOI: https://doi.org/10.7554/eLife.44707.005

## Structure at the AAV2-PKD2 binding interface

On receptor-binding, AAV2 undergoes only limited conformational change. At variable region (VR)-I, residues 263–266 are displaced 2.1–3.9 Å (C$_\alpha$) and become more disordered through conflict with AAVR residues 123–125. The high-resolution structure shows no evidence of longer range conformational change. (A caveat should be noted, that, even if there is little change in the static or average structure of the major capsid protein, it has been reported that capsid stability decreases when DNA content is increased (*Horowitz et al., 2013*). This may be relevant because the virus-like particles, VLPs, do not have the same nucleic acid content as wild-type virus or vectors.) The footprint of PKD2

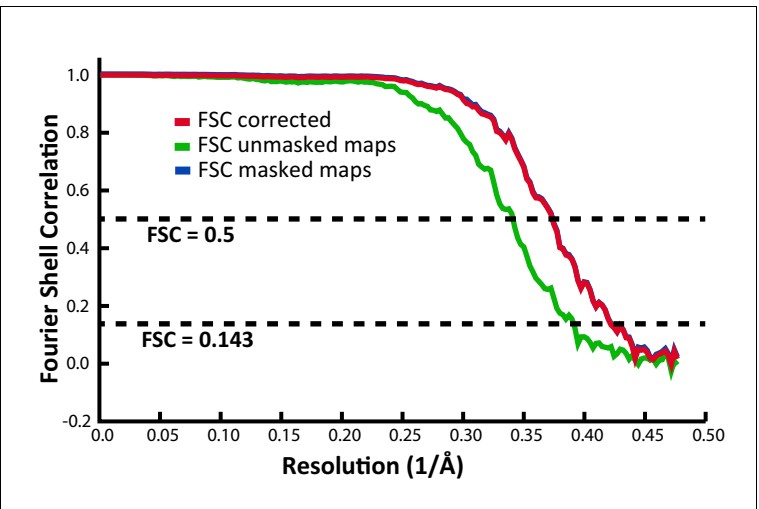

**Figure 5.** Fourier shell correlation (FSC) indicating an overall resolution of 2.39 Å.
DOI: https://doi.org/10.7554/eLife.44707.006

on AAV2 straddles the plateau that extends radially from each threefold axis, beyond two adjacent spikes (*Figure 6*). PKD2's N-terminus is near the twofold. The domain body passes along the plateau, below the nearest spike, and its C-terminus rises towards the shoulder of the next-nearest spike (*Figure 6*).

## Comparison of the AAVR binding site to those of neutralizing antibodies

Comparison of the receptor-binding site with neutralizing epitopes is of both fundamental and applied interest. Fundamentally, it could be informative on the mechanisms by which antibodies reduce infectivity, a topic on which there is some experimental data, but speculation continues (*Wobus et al., 2000*; *Harbison et al., 2012*; *Gurda et al., 2013*). In application of AAV vectors for gene therapy, neutralization and/or clearance, and even tissue retargeting in the presence of neutralizing antibodies (NAbs) are practical challenges in the safety and efficacy of treatments in development (*Manno et al., 2006*; *Wang et al., 2010*; *Corden et al., 2017*; *Fitzpatrick et al., 2018*). This is not just an issue with immune-sensitization of patients when multi-dose treatment regimens are envisioned, but ~60% of the population are seropositive to various AAV serotypes, due to natural exposure (*Calcedo et al., 2009*; *Boutin et al., 2010*).

**Table 1.** Structural analysis of virus-receptor complexes cross-linked with CBDPS at sites identified by tandem mass spectrometry following proteolytic digestion and affinity purification of cross-linked peptides.

The spacer length of CBDPS is 14 Å. Distances are measured from models based on electron microscopy of complexes that were not cross-linked. Thus, the distances do not reflect any chemical constraint imposed by the cross-linker on conformation, or any remodeling of the protein structure.

| Virus | Residue | AAVR construct | Residue | Location | Distance | Measured from: |
|-------|---------|----------------|---------|----------|----------|----------------|
| AAVDJ | K556 | Full *ecto*-protein | K404 | PKD2 (N-terminal) | 13.4 Å | PKD2 modeled into high resolution EM of AAV2-PKD1/2 |
| AAV2 | K490 | MBP-PKD1-5 | K399 | PKD1/2 linker | 9.9 Å | PKD1 homology model, anchored to PKD2 in above EM |
| AAV2 | T560 | MBP-PKD1-5 | K399 | PKD1/2 linker | 17.8 Å | As above |
| AAV2 | K556 | MBP-PKD1-5 | K338 | PKD1 | 13.8 Å | As above |
| AAV2 | T450 | MBP-PKD1-5 | K399 | PKD1/2 linker | 27.7 Å | As above |
| AAV2 | K556 | MBP-PKD1-5 | K597 | PKD3/4 linker | | Not attempted |

DOI: https://doi.org/10.7554/eLife.44707.007

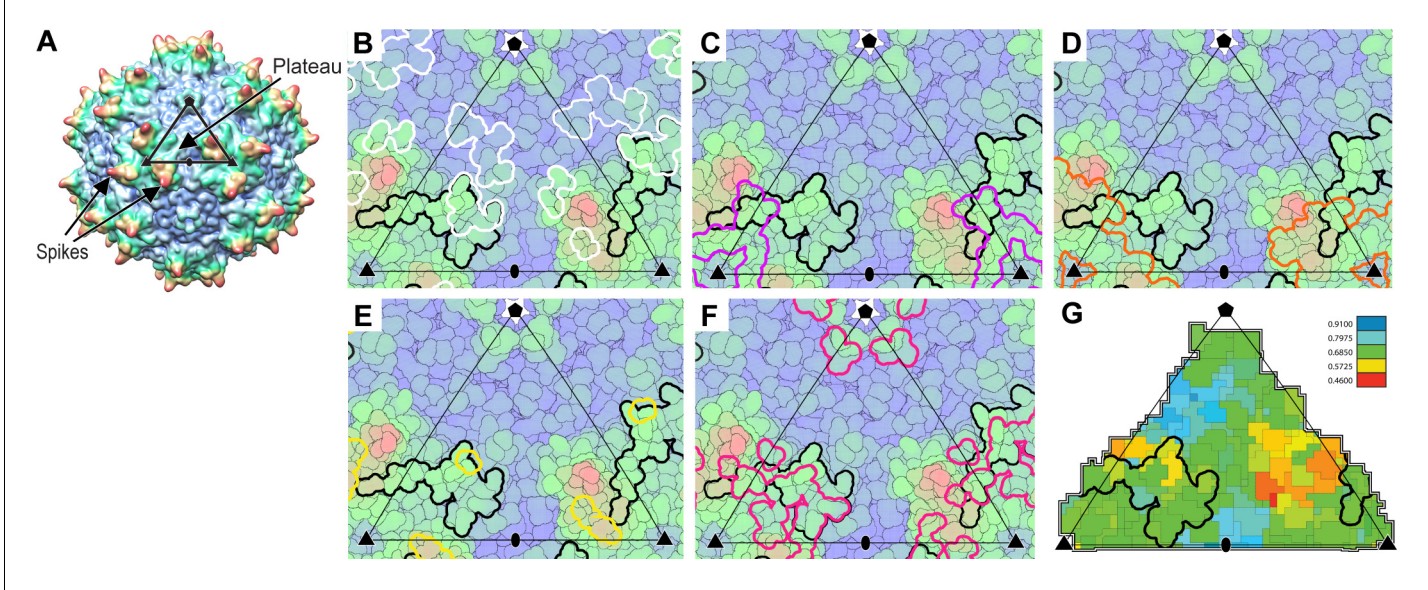

**Figure 6.** Interactions of AAVR with AAV. (**A**) The surface of AAV can be divided into 60 equivalent triangular asymmetric units bounded by a 5-fold and two 3-fold symmetry axes, and containing parts of several subunits together adding to one. In panels **A-E**, the viral surface is colored blue-to-red by distance from the virus center. (**B-E**) Overlaid on an asymmetric unit are outlined the footprints of AAVR-PKD2 (black; *Table 2*) and neutralizing monoclonal antibodies: A20 (**B**; white), C37B (**C**; purple), D3 (**D**; orange) and C24B (**E**; yellow) (*Wobus et al., 2000*; *McCraw et al., 2012*; *Gurda et al., 2013*). (Residues are labeled in the supplement.) (**F**) The sites of 'dead zone' transduction-abrogated mutations are outlined in pink (*Lochrie et al., 2006*). (**G**) The AAVR footprint is outlined over a projection colored by sequence identity (among presumptive AAVR-binding serotypes) from conserved blue (>90% identity) to variable red (<50% identity).
DOI: https://doi.org/10.7554/eLife.44707.008

The following figure supplement is available for figure 6:

**Figure supplement 1.** Overlap between the AAVR footprint on AAV2 and epitopes of neutralizing monoclonal antibodies.
DOI: https://doi.org/10.7554/eLife.44707.009

The epitope of the best characterized neutralizing monoclonal antibody (MAb) A20 (*McCraw et al., 2012*) overlaps with the AAVR footprint, and suggests that entry or trafficking might be blocked even without inhibition of (glycan-mediated) cell attachment (*Wobus et al., 2000*) (*Figure 6B*). Overlap is seen for the other AAV2-neutralizing monoclonal antibodies (MAbs; *Figure 6C–E*). Clashes with PKD2 or adjacent domains are predicted when the PKD2-AAV2 footprint is overlaid on MAb complexes of AAV1/6 (*Gurda et al., 2013*; *Tseng and Agbandje-McKenna, 2014*). Neutralization is not as well predicted by receptor-epitope overlap for AAV5 and AAV8, but it is premature to imply different neutralization mechanisms: (a) Lower resolution structures of other

**Table 2.** Amino acids at the AAV2-AAVR interface.
Residues of AAVR are listed if any non-hydrogen atom is within 4.5 Å of any non-hydrogen AAV2 atom, et vice versa. This criterion corresponds approximately to the distance expected between methyl groups that are in van der Waals contact, and is intermediate between that of hydrogen-bonding and solvent exclusion. However, to assess the potential for specific interactions, readers are encouraged to inspect the deposited coordinates and maps (PDBid 6NZ0/EMD-0553).

**AAVR residues close to AAV2:**

| | | | | | | | | | | | |
|---|---|---|---|---|---|---|---|---|---|---|---|
| $Arg_{406}$ | $Ser_{413}$ | $Ile_{419}$ | $Thr_{423}$ | $Ser_{425}$ | $Thr_{426}$ | $Val_{427}$ | $Asp_{429}$ | $Ser_{431}$ | $Gln_{432}$ | $Ser_{433}$ | $Thr_{434}$ |
| $Asp_{435}$ | $Asp_{436}$ | $Asp_{437}$ | $Lys_{438}$ | $Ile_{439}$ | $Tyr_{442}$ | $Glu_{458}$ | $Asp_{459}$ | $Ile_{462}$ | $Lys_{464}$ | | |

**AAV2 residues close to AAVR:**

| | | | | | | | | | | | |
|---|---|---|---|---|---|---|---|---|---|---|---|
| $Gln_{263}$ | $Ser_{264}$ | $Gly_{265}$ | $Ala_{266}$ | $Ser_{267}$ | $Asn_{268}$ | $His_{271}$ | $Asn_{382}$ | $Gly_{383}$ | $Ser_{384}$ | $Gln_{385}$ | |
| $Arg_{471}$ | $Trp_{502}$ | $Thr_{503}$ | $Asp_{528}$ | $Asp_{529}$ | $Gln_{589}$ | $Lys_{706}$ | $Val_{708}$ | | | | |

DOI: https://doi.org/10.7554/eLife.44707.010

MAb complexes, (*Gurda et al., 2013*; *Tseng and Agbandje-McKenna, 2014*), and inaccuracies in using AAV2 to predict the exact PKD2 footprints of other serotypes, limit assessment of overlap; (b) From AAV5, we know that interactions with additional PKD domains can be important (*Pillay et al., 2017*), likely extending the footprint beyond that seen in the AAV2 structure; and (c) Antibody-receptor conflict might not be limited to the visualized footprint of PKD2, but could, in principle, occur between other domains, even distal from the viral surface, noting that AAVR and many neutralizing antibodies are anchored by binding sites in the same general vicinity around the threefold axes. In summary, it is plausible that the neutralization mechanism of AAV antibodies might commonly be steric blocking of AAVR interactions, although other mechanisms are likely also at work.

## Conservation of the AAVR binding site

Importance of the PKD2 footprint to AAV is evidenced in the striking correspondence to a 'dead-zone', where substitution mutations abrogate transduction (*Figure 6F*) (*Lochrie et al., 2006*). Excluding the AAV4 clade, which, alone, does not bind AAVR, sequence is less variable than on other antibody-accessible surfaces, suggesting a greater evolutionary/functional cost to immune-escape mutation (*Figure 6G*). Sequence-variable regions (VR) were previously defined by differences between the AAV2 and AAV4 structures (*Govindasamy et al., 2006*). VR-I and III are at the core of the AAVR footprint. Together with the fivefold proximal VRII, sequence variability is actually markedly less than in VR IV-IX, specifically among the AAVR-binding serotypes that exclude the AAV4 clade (*Figure 7*). Structural conservation of the PKD2 footprint does not extend to AAV4 and AAVrh32.33 which differ at the two principal contact points, VR-I and III (*Figure 8*). Superimposition on the AAV2-AAVR complex shows binding-incompatibility of the AAV4 clade, due to sequence insertion and deletion in the two loops, explaining the recent finding that AAV4, alone among representative extant and ancestral primate AAVs, uses a different receptor (*Dudek et al., 2018*).

## Juxtaposition of the AAVR receptor-binding site with glycan attachment sites

AAV's attachment to cells is mediated through extracellular glycans, bound at sites characterized for several serotypes through structure, or implied from mutations affecting attachment or cell entry (*Figure 9*) (*Huang et al., 2014*). AAVR is glycosylated at $Asn_{472}$ and $Asn_{487}$, but these sites face away from the virus in our structure, and their glycosylation is not required for viral entry (*Pillay et al., 2017*), so it is non-AAVR glycans that mediate attachment. When our structure is superimposed on those of AAV-glycan complexes, AAVR conflicts with heparan sulfate analogs at the site shared by AAV2 and AAVDJ (*O'Donnell et al., 2009*; *Xie et al., 2017*; *Xie et al., 2013*). $Arg_{585}$ and $Arg_{587}$ (AAV2 numbering) interact electrostatically either with glycan sulfates or AAVR $Asp_{459}$. There is some overlap between AAVR and AAV1/6-bound sialic acid (*Huang et al., 2016*), but none with sucrose octasulfate bound to AAV3B at $Arg_{594}$ (*Lerch and Chapman, 2012*). The viral symmetry would allow some sites on AAV to remain glycan-attached, while others become AAVR-bound. The inter-domain flexibility that complicated structure determination might be key, in vivo, allowing cell-distal domains of AAVR to be bound at open sites on AAV.

## Discussion

### Hybrid methods

This investigation establishes EM as a centerpiece in the structure determination of flexible molecules, using hybrid-approach divide-and-conquer strategies. Resolution was limited in complexes with PKD1-5 constructs because of: (a) heterogeneity in domain orientations of a hinged receptor; (b) occupancy reduced by conflicts, not between PKD2 domains themselves at neighboring symmetry-related viral sites, but downstream domains (PKD3-5); and likely (c) oligomerization of larger receptor constructs, which, when well separated-particles are selected from EM images, biases processing toward virions with lower receptor occupancy. Redundancy in approaches allowed the project to advance beyond limitations that came into perspective only in retrospect.

The tomographic reconstruction at ~30 Å resolution lacked the detail needed to determine which receptor domains were virus-proximal. Cross-linking/mass spectrometry (x-MS) identified contact regions in primary sequence, determining overall receptor orientation. It allowed crude modeling of

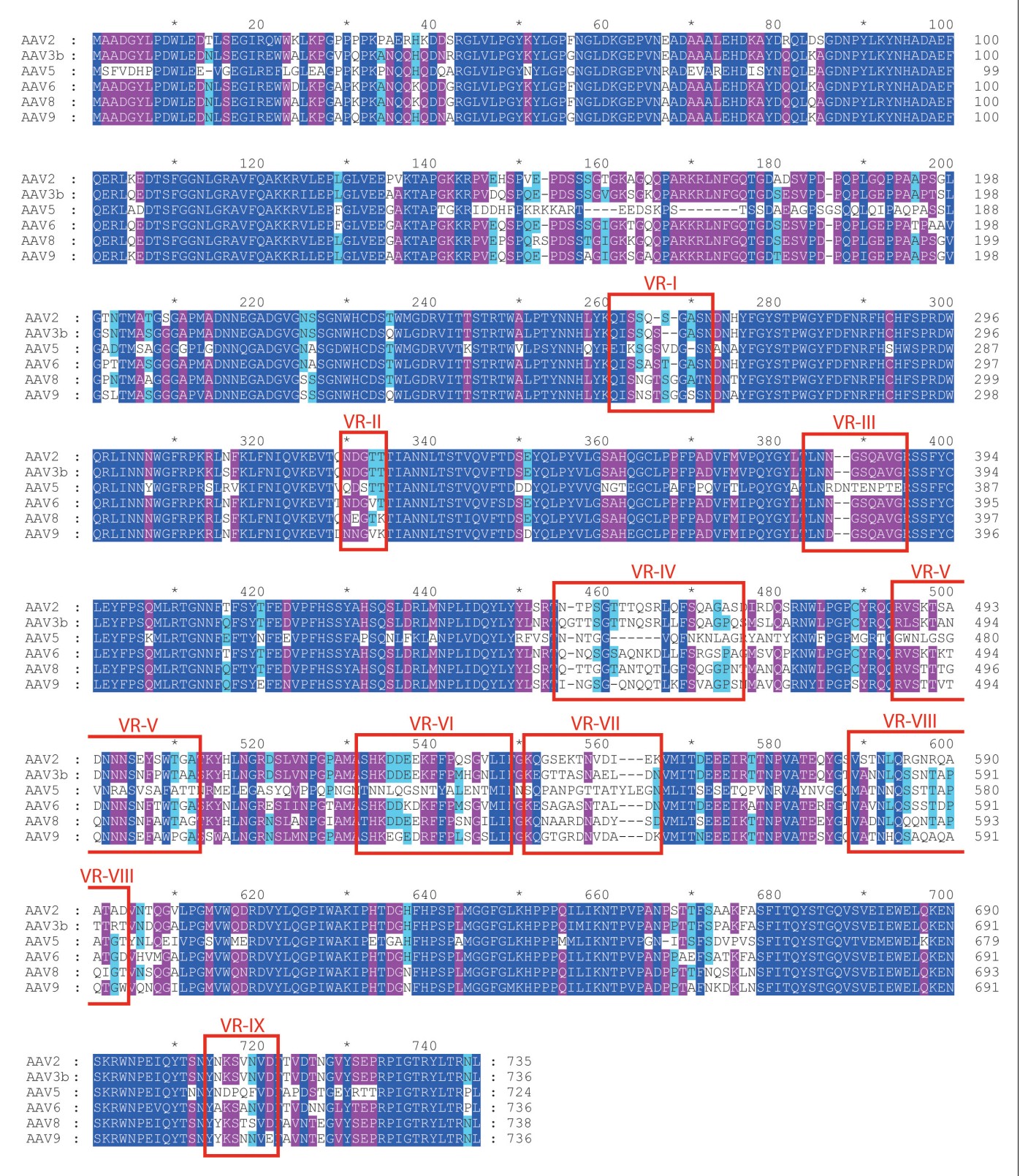

**Figure 7.** Aligned AAV sequences, highlighting variable regions (VR) I to IX. Representative sequences were aligned using Clustal (*Chenna et al., 2003*). Boxes highlight the regions designated as variable when the atomic structure of AAV4 was compared to AAV2 (*Govindasamy et al., 2006*). For VR IV through IX, sequence diversity extends through all strains. However, variability in VR I through III arises primarily through divergence of the AAV4

*Figure 7 continued on next page*

*Figure 7 continued*

clade from others. VR-I and VR-III constitute the surface loops with most intimate interactions between AAV2 and AAVR, that is excluding the AAV4 clade, variability is not elevated in VR-I and VR-III compared to other surface regions.

DOI: https://doi.org/10.7554/eLife.44707.011

the tomographic density through the addition of x-MS distance constraints. Ambiguities remained, because, with allowance for protein flexibility, cross-linking to symmetry-related subunits was possible. Absent symmetry, the combination of *cryo*-ET and x-MS would have allowed unambiguous domain-level *pseudo*-atomic modeling. Furthermore, with hindsight of the high-resolution structure, distance constraints could have been applied more stringently, because cross-links were formed with little adaptation of protein structure. This work provides an affirmation that low resolution electron microscopy and x-MS can be a powerful combination, even with flexible molecules.

High resolution came using a small receptor fragment, but it is the low-resolution EM and x-MS that establishes the biological relevance of the construct. Concordance of the PKD2 structure at 2.4 Å resolution with the ~10 Å view of PKD2 from SPA/sub-volume extraction and the ~30 Å *cryo*-ET, both from complexes with the entire PKD1-5, argues that the atomic interactions revealed at 2.4 Å are not an artifact of using a truncated PKD1-2 receptor, as does consistency of the atomic model with distances expected from x-MS with the five-domain construct or a near-complete receptor *ecto*-domain.

## Antibody interference with AAVR-binding

One surprise from our structure might be the overlap between the PKD2 footprint and epitopes for some of the AAV-neutralizing monoclonal antibodies, given prior statements about particular

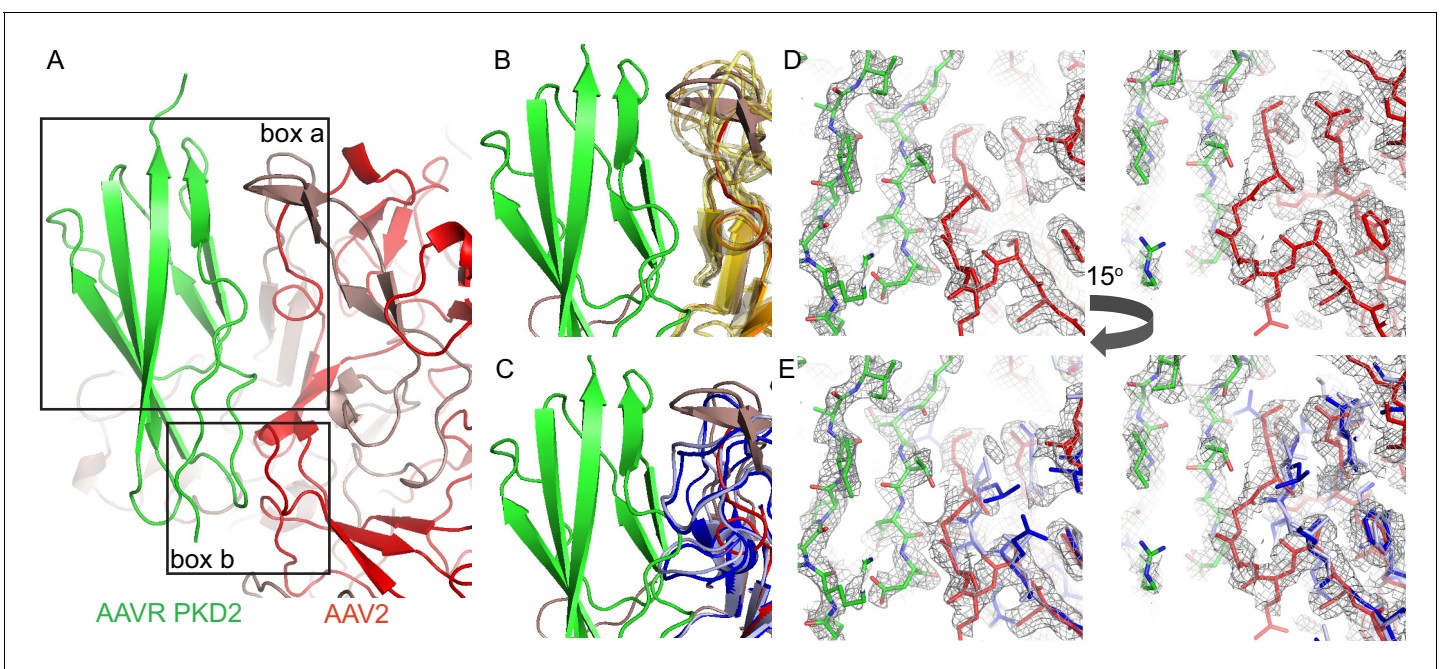

**Figure 8.** Interactions between AAVR and AAV-2. (**A**) Boxed areas show contact points between PKD2 of AAVR (green) and two adjacent subunits of AAV2 (red). (**B**) Boxed volume 'a', now with AAV serotypes 3b, 5, 6, 8, 9, and DJ superimposed (translucent, yellow to brown) (*Lerch et al., 2010*; *Walters et al., 2004*; *Xie et al., 2011*; *Nam et al., 2007*; *DiMattia et al., 2012*; *Xie et al., 2017*), all of which can accommodate AAVR. (**C**) The same volume 'a', superimposing AAV4 (dark blue) and AAVrh32.33 (light blue) (*Govindasamy et al., 2006*; *Mikals et al., 2014*) that have a loop insertion in variable region VRIII of the sequence that would clash with AAVR. (**D** and **E**) EM density for the AAV2 complex in boxed volume 'b', contrasting the complementarity with AAVR for AAV2 (**D**, red) with poor complementarity (**E**), due to a deletion (variable region VRI of the sequence), in the superimposed crystal structures of AAV4 (dark blue) and AAVrh32.33 (light blue).

DOI: https://doi.org/10.7554/eLife.44707.012

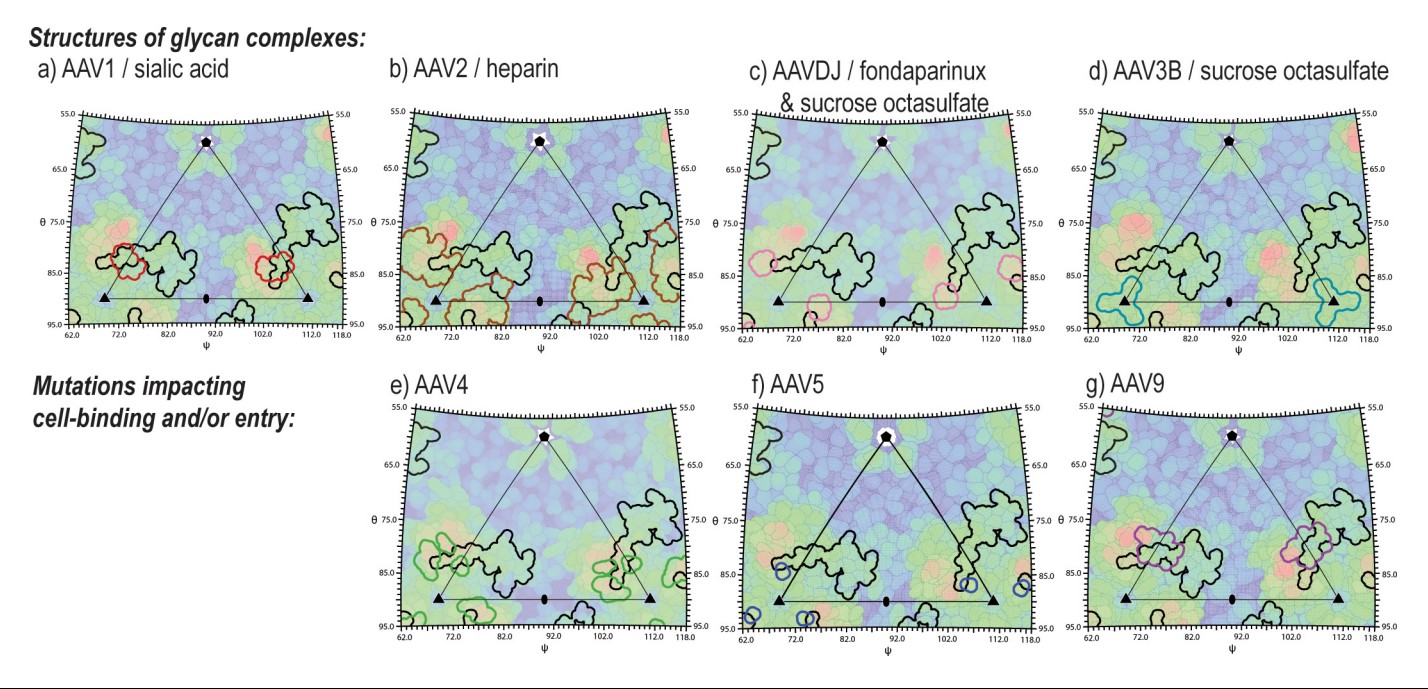

**Figure 9.** Juxtaposition of AAV glycan attachment sites, and the AAV2 contact footprint of entry receptor, AAVR. Panels show the surface topologies of different AAV serotypes, projected as in *Figure 6*, colored, blue to red, with increasing distance from the virus center. The contact footprint of AAVR on AAV2 overlaid is outlined in black. Outlined in color are amino acids that contact glycan analogs in structures of complexes (AAV1-sialic acid [*Huang et al., 2016*], AAV2-heparin [*O'Donnell et al., 2009*], AAV3B with sucrose octasulfate [*Lerch and Chapman, 2012*], AAV-DJ with sucrose octasulfate [*Xie et al., 2013*] or fondaparinux [*Xie et al., 2017*]) or that have been implicated by mutagenesis in cell attachment and/or uptake (AAV4 [*Shen et al., 2013*], AAV5 [*Afione et al., 2015*] and AAV9 [*Bell et al., 2012*]).
DOI: https://doi.org/10.7554/eLife.44707.013

antibodies not impacting receptor-binding (eg. *Gurda et al., 2012*). While our findings contradict some prior conclusions, they are actually consistent with much of the underlying data. Confusion has resulted from the lack of distinction in the historical literature between the attachment of AAV to cells and its entry. This distinction would not have been important to the current discussion should the initial interactions with extracellular glycans, termed 'primary receptors', still be considered the pivotal step in cell entry. However, it is now known that they impact transduction much less than the AAVR protein (*Pillay et al., 2016*) which has more of the properties of a classical entry receptor. From the virus' perspective, the glycans should not be considered as receptors, but as cell-attachment factors whose interactions may not be highly specific (*Zhang et al., 2013*; *Pillay et al., 2016*). Past discussions of neutralization mechanisms have included measurements of (glycan-dominated) cell-binding which tell us whether neutralization might be pre- or post-attachment, but not about interference with entry receptors, or whether neutralization is post-entry (*Harbison et al., 2012*; *Gurda et al., 2012*; *Gurda et al., 2013*; *Pillay et al., 2016*).

Of three neutralizing anti-AAV monoclonals whose mechanisms have been investigated, two (A20 and ADK8) were deemed to be post-entry (*Wobus et al., 2000*; *Gurda et al., 2012*). It might have appeared counter-intuitive that we now find overlap of the AAVR-binding site with neutralizing epitopes, but not with the current understanding of AAV entry. While glycans dominate surface attachment, AAVR is needed in endosomal trafficking of AAV from the cell surface toward the nucleus for productive infection/transduction (*Pillay et al., 2016*). More care will be needed to differentiate potential steps of antibody inhibition. We now know that glycan attachment and endosomal churning can yield positive immunofluorescence indicating internalization, even though we know, from knock-outs, that that AAVR is needed for productive transduction (*Pillay et al., 2016*). Not only do we need to distinguish extracellular attachment and cellular internalization, but we need to differentiate virions that undergo receptor-mediated trafficking and endosomal escape, from unproductive

virions that get no further than the endosome. To reiterate in other words, future analyses should reflect our new understanding that the glycans, until recently considered primary receptors, are more properly considered as attachment factors, and that productive entry (from the endosome into the cytoplasm) depends on interactions with different protein receptors, like AAVR. The current work shows that binding of some of the most neutralizing MAbs interferes with these interactions. However, we note that diversity in antibody-binding sites has previously been noted, and that a variety of neutralization mechanisms, not just inhibition of AAVR binding, will likely be in play for different antibodies (*Gurda et al., 2013*).

### Future prospects

The structure of the AAV-receptor complex opens new chapters both in realizing the potential for gene therapy to treat an array of genetic diseases, and, it is hoped, in fundamental virology. The structure provides a roadmap that will support experimental perturbation of the interactions, a step toward rational modulation of cell targeting and neutralization escape (*Asokan et al., 2012*). Relevant to engineering vectors to escape immune neutralization from pre-existing NAbs, the current work highlights the overlap/juxtaposition of epitopes to be targeted, with the binding site for AAVR, the integrity of which is needed for productive infection/transduction. This work will be a key foundation for attempts to modulate immune interactions without collateral disruption of cell entry. More ambitious would be attempts to improve the efficiency or specificity of vectors by adding functionality to the AAVR-binding site. However, the drive for clinical impact will be strong motivation for gain-of-function studies at an unprecedented level, so this work opens the door to AAV becoming a particularly valued structure-function model for the fundamentals of viral-host interactions.

### Note added in proof

During review, a structure showing AAVR's PKD2 bound to AAV2 at 2.8 Å resolution became available (*Zhang et al., 2019*). Mostly the structures validate each other. However, while *Zhang et al. (2019)* found the sites to be non-overlapping, we find overlap between the binding site of AAVR and neutralizing monoclonal antibody A20 (*McCraw et al., 2012*), and with analogs of heparan sulfate (*Xie et al., 2013*; *Xie et al., 2017*). (Our residue numbering starts from the N-terminus of wild-type AAVR and corresponds if 260 is added to the PKD2 model of *Zhang et al., 2019*.)

## Materials and methods

**Key resources table**

| Reagent type (species) or resource | Designation | Source or reference | Identifiers | Additional information |
|---|---|---|---|---|
| Strain, strain background (*Escherichia coli*) | BL21(DE3) *E. coli* | ThermoFisher | ThermoFisher: C601003 | |
| Strain, strain background (*E. coli*) | NEB Express *E. coli* | New England Biolabs | NEB: C2523I | |
| Cell line (*Spodoptera frugiperda*) | Sf9 | Gibco | Gibco: 11496015 | |
| Recombinant DNA reagent | pET-11a | Novagen | EMD Milli-pore: 69436–3 | |
| Recombinant DNA reagent | pMAL-c5X | New England Biolabs | NEB: N8108S | |
| Commercial assay or kit | Cleavable ICAT Reagent Kit for Protein Labeling | SCIEX | | |

*Continued on next page*

Continued

| Reagent type (species) or resource | Designation | Source or reference | Identifiers | Additional information |
|---|---|---|---|---|
| Commercial assay or kit | Bac-to-Bac Baculovirus Expression System | ThermoFisher (Invitrogen) | ThermoFisher: 10359016 | |
| Commercial assay or kit | MBPTrap HP column | GE | GE: 28918778 | |
| Commercial assay or kit | HiTrap Chelating HP column | GE | GE: 17040801 | |
| Commercial assay or kit | Superdex 75/200 column | GE | GE: GE17-5174-01 | |
| Software, algorithm | Leginon | *Suloway et al., 2005* doi: 10.1016/j.jsb.2005.03.010 | RRID:SCR_016731 | |
| Software, algorithm | Protomo | *Winkler, 2007* doi: 10.1016/j.jsb.2006.07.014 | | |
| Software, algorithm | TOMOCTF | *Fernández et al., 2006* doi: 10.1016/j.jsb.2006.07.014 | | |
| Software, algorithm | Dynamo | *Castaño-Díez et al., 2012* doi: 10.1107/52059798317003369 | | |
| Software, algorithm | Relion 3.0 | *Scheres, 2012* doi: 10.1016/j.jsb.2012.09.006 | RRID:SCR_016274 | |
| Software algorithm | Motioncor2 1.1.0 | *Zheng et al., 2017* doi: 10.1038/nmeth.4193 | RRID:SCR_016499 | |
| Software, algorithm | Gctf 1.06 | *Zhang, 2016* doi: 10.1016/j.jsb.2015.11.003 | RRID:SCR_016500 | |
| Software, algorithm | Localized reconstruction | *Ilca et al., 2015* doi: 10.1038/ncomms9843 | | |
| Software, algorithm | StavroX | *Götze et al., 2012* doi: 10.1007/s13361-011-0261-2 | RRID:SCR_014957 | |
| Software, algorithm | Coot | *Brown et al., 2015* doi: 10.1017/S1399004714021683 | RRID:SCR_014222 | |
| Software, algorithm | MapMan | | RRID:SCR_003543 | |
| Software, algorithm | EMAN | | RRID:SCR_016867 | |
| Software, algorithm | RSRef | *Chapman et al., 2013* doi: 10.1016/j.jsb.2013.01.003 | RRID:SCR_017211 | |
| Software, algorithm | Roadmap | *Chapman, 1993* doi: 10.1002/pro.5560020318 | RRID:SCR_017207 | |
| Software, algorithm | Rivem | *Xiao and Rossmann, 2007* doi: 10.1016/j.jsb.2006.10.013 | | |

*Continued on next page*

*Continued*

| Reagent type (species) or resource | Designation | Source or reference | Identifiers | Additional information |
|---|---|---|---|---|
| Software, algorithm | Modeller 9.2 | *Eswar et al., 2006* doi: 10.1007/978-1-60327-058-8_8 | RRID:SCR_008395 | |
| Software, algorithm | Chimera | | RRID:SCR_002959 | |
| Software, algorithm | Pymol | | RRID:SCR_000305 | |

## Virus and receptor preparation

AAV2 virus like particles (VLPs) were expressed in Sf9 cells using Invitrogen's Bac-to-Bac expression system (*Urabe et al., 2002*). Empty capsids were purified using three rounds of CsCl density gradient ultracentrifugation, followed by heparin affinity chromatography, eluting with NaCl. Capsids were then dialyzed in 25 mM HEPES, 50 mM $MgCl_2$, 150 mM NaCl, pH 7.4. PKD domains 1–5 of AAVR were expressed in BL21(DE3) *E. coli* using the pMAL expression system (New England Biolabs). This construct (MBP-PKD1-5) comprised a maltose-binding protein (MBP) tag fused N-terminally to the PKD domains. cDNA coding for AAVR PKD domains 1–5 was cloned into the pMAL-c5X expression vector and expression was carried out in NEB Express *E. coli* cells (New England Biolabs). Fusion protein was purified chromatographically using an MBPTrap HP column followed by a HiTrap Chelating HP column (GE) charged with $Co^{2+}$. AAVR constructs comprising PKD1-2 were expressed with an N-terminal 6x-histidine tag from the pET-11a vector (Novagen) and were purified by immobilized $Co^{2+}$ affinity followed by size exclusion chromatography (Superdex 75/200, GE).

## *Cryo*-tomography

MBP-PKD1-5 and AAV2 VLP were incubated briefly together at a molar ratio of 54:1 (AAVR:AAV2-subunit). Quantifoil R1.2/1.3 grids (Electron Microscopy Sciences) were glow discharged at 15 mA for 25 s (Pelco easiGLOW), prior to 2-min incubation with 2.5 µl of complex, subsequent wicking, and addition of 4.5 µl of complex before blotting and plunge-freezing using a Vitrobot Mark IV (FEI). *Cryo*-ET tilt series were acquired on an FEI Titan Krios (FEI, Hillsboro, OR) and recorded with Leginon software (*Suloway et al., 2009*) on a DE-20 direct detector (Direct Electron, San Diego, CA). An exposure magnification of 18,000 was used with a nominal pixel size of 2.03 Å. Total dose was 50 e–/$Å^2$ per tilt series. The tilt scheme involved rotation from −45° to 60° in 3° steps, and the dose was fractionated across seven frames at each step. Defocus values were set to range from 9 µm to 11 µm. Fractionated frame images at each tilt angle were motion-corrected using an open-source python script (DE_process_frames.py) provided by Direct Electron. Tilt series were then aligned using Protomo software within Appion (*Winkler and Taylor, 2006*; *Lander et al., 2009*; *Noble and Stagg, 2015*). CTF estimation and correction were performed using TOMOCTF (*Fernández et al., 2006*) and tomograms reconstructed with Tomo3D WBP (*Agulleiro and Fernandez, 2015*). The averaged power spectrum of $1024^2$ pixel regions was calculated and average defocus values were re-estimated for the eight tilt series, using TOMOCTF, yielding a new range of 9.7–11.5 µm. Phase-flipped CTF correction was then applied using stripes of 1000 Å, followed by dose compensation using parameters output by Appion-Protomo. In total, eight 3D tomographic maps were then reconstructed from the above image stacks using Tomo3D.

For subtomogram picking, a 50 Å low-pass filtered map of uncomplexed AAVDJ (EMD6470) (*Xie et al., 2017*) was used. MolMatch was used to calculate a constrained correlation coefficient map (*Förster et al., 2010*), and an in-house automated template picking program was used to pick a total of 1321 AAV-2 particles. Picked subtomograms were extracted in Dynamo (*Castaño-Díez et al., 2012*) prior to alignment. The reference was the AAVDJ map, low pass filtered at 60 Å resolution to avoid model bias. A coarse search with fourfold binning was followed by a finer (0.5°) refinement with twofold binning and with icosahedral symmetry imposed. Aligned capsid subvolumes were rotated into a standard reference frame defined by icosahedral axes, then symmetry-expanded to generate 60 redundant copies. When viral particles are overlaid, a fraction, classed by manual inspection, had receptor, usually one, bound near a given threefold. Sub-volumes containing

**Table 3.** Cryo-EM AAV2-PKD1-2 data collection and processing statistics.

| Data collection: | |
|---|---|
| Magnification | 75,000 x |
| Voltage | 300 kV |
| Electron exposure | 25 e/Å$^2$ |
| Defocus range | −0.8 to −2.0 μm |
| Pixel size | 1.049 Å |
| **Data processing:** | |
| Motion correction | Motioncor2 1.1.0 |
| *Anisotropic magnification correction:* | |
| Distortion angle | 3.2˚ |
| Percent distortion | 1.10% |
| CTF estimation | Gctf 1.06 |
| Resolution range | 30 to 3 Å |
| Symmetry imposed | I1 |
| Initial particle images | 34,450 |
| Final particle images | 21,373 |
| Map resolution | 2.39 Å |
| FSC threshold | 0.143 |

DOI: https://doi.org/10.7554/eLife.44707.014

one of the three spikes surrounding the symmetry axis, about a third containing receptor, were classified and averaged using the I3 package (*Hu et al., 2011*; *Winkler et al., 2009*; *Winkler and Taylor, 1999*).

## Single-particle *cryo*-EM

The MBP-PKD1-5 complex for cryo-EM was prepared as for cryo-ET. Data were acquired on a Titan Krios using Leginon (*Suloway et al., 2005*) on a DE-20 detector. Magnification was 29,000, pixel size was 1.256 Å, and defocus range was set to -1.5 to -3.0 μm. Total dose was ~69 e-/Å$^2$ per image. 2D and 3D classification in Relion 1.4 (*Scheres, 2012*) culled the number of selected particles to 36208, and 3D auto-refinement resulted in a final unmasked map at an overall resolution of 4.2 Å. Subsequent localized classification of individual threefold spikes allowed reconstruction of those spikes at which receptor density was present (*Ilca et al., 2015*).

For the PKD1-2 complex, grids coated with ultrathin carbon over lacey carbon (Ted Pella Cat No 01824) were glow discharged at 25 mA for 25 s (Pelco easiGLOW). 4 μl of AAV2 (1.7 μM VP subunits) was added to the grid, followed by 4 μl of PKD1-2 (16.7 μM) in buffer HN (25 mM HEPES, 150mM NaCl, pH 7.4), with blotting between and after (Whatman Cat No 1001–110). 4 μl of HN buffer was added to the grid before final blotting and plunge-freezing. Single-particle data with PKD1-2 were collected on a Titan Krios with a Falcon three direct detector (FEI) using EPU software (FEI). Pixel size was 1.049 Å and each movie contained 160 frames with a total dose of 25 e-/Å$^2$/ movie. Defocus ranged from -0.8 μm to -2.0 μm. A total of 2329 movies were motion corrected using MotionCor2 1.1.0 (*Zheng et al., 2017*) and initial CTF estimation of non-dose-weighted images was done using Gctf 1.06 (*Zhang, 2016*). Processing continued within RELION 3.0: 34,450 particles were initially picked with AutoPicker using a 3D AAV2 reference filtered to 20 Å with 15˚ sampling. Three rounds of 2D classification (resampled to ~2.1 Å/pixel) followed by two subsequent rounds of 3D classification (K = 2) culled the number of particles to 33,555. Initial 3D auto-refinements, unmasked then masked, imposed icosahedral symmetry (I1). Two further rounds of CTF refinement followed by auto-refinements preceded an additional 3D alignment-free classification (K = 2) which resulted in a dataset of 21,373 particles. Final masked 3D auto-refinement imposing I1 symmetry resulted in a map of 2.39 Å resolution (FSC gold standard [*Zivanov et al., 2018*]). Further details are presented in *Table 3*.

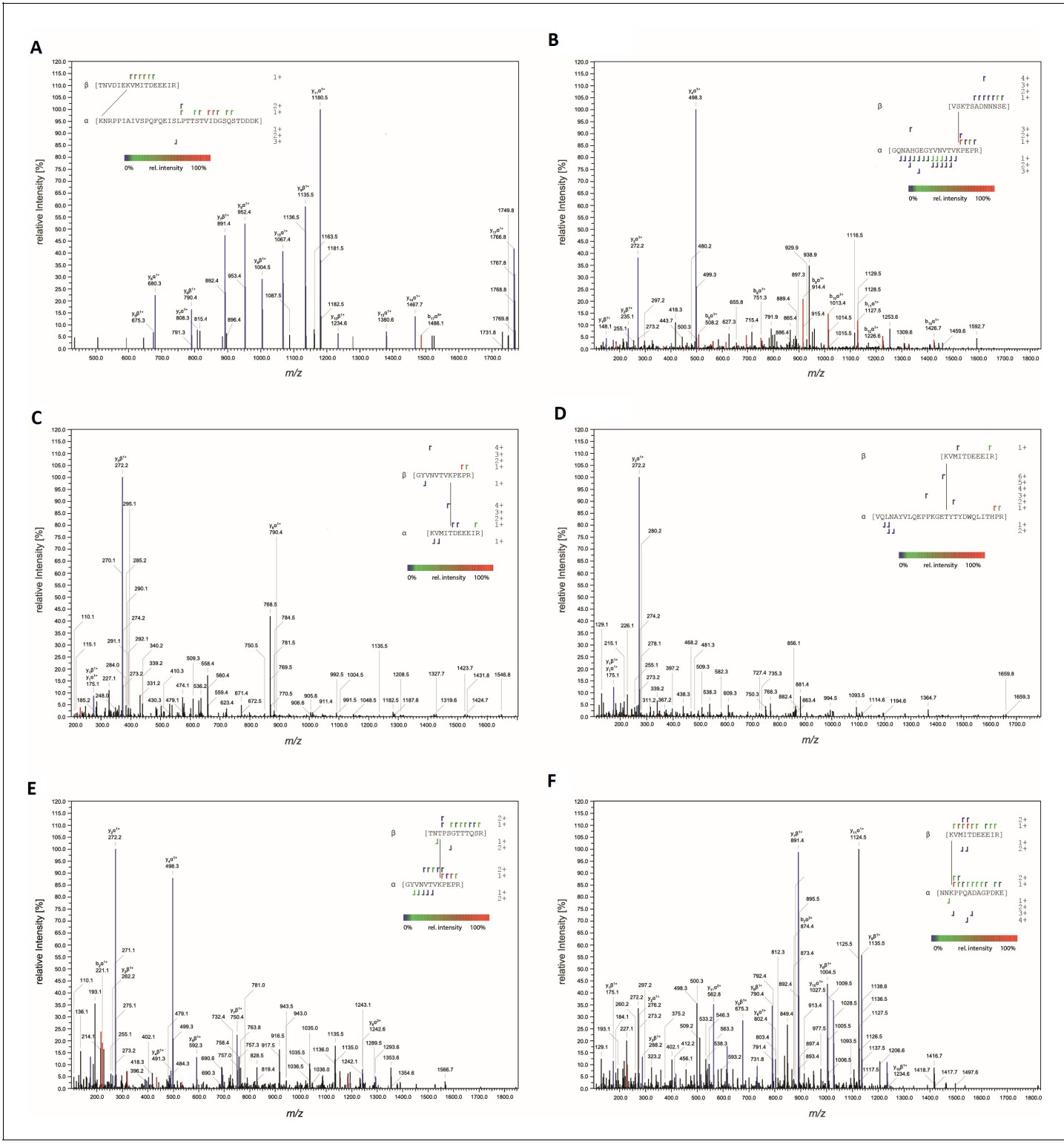

**Figure 10.** MS-MS characterization of cross-linked peptides. Fragmentation spectra of peptides cross-linked at (**A**) AAVDJ K556:AAVR K93, (**B**) AAV2 K490:AAVR K88, (**C**) AAV2 T560:AAVR K88, (**D**) AAV2 K556:AAVR K27, (**E**) AAV2 T450:AAVR K88, and (**F**) AAV2 K556:AAVR K286. Peaks corresponding to y and b fragments are colored blue and red, respectively. Resulting fragment ions shown in the insets.
DOI: https://doi.org/10.7554/eLife.44707.015

## Modeling of PKD2

One PKD domain could be traced readily in the reconstruction. Superimposition of a canonical seven-stranded immunoglobulin-like domain gave an unambiguous orientation with the N-terminal end closer to the viral twofold and the C-terminal end proceeding toward the threefold axis. Homology modeling (*Eswar et al., 2006*) provided a starting point for model-building with Coot (*Brown et al., 2015*), iterated with the refinement of the model and imaging parameters using RSRef (*Chapman et al., 2013*). The latter started with rigid fitting of the receptor domain, and progressed through refinement of the EM magnification, EM envelope, effective resolution, receptor occupancy and B-factor relative to virus, searches for best-fitting library side-chain rotamers (*Lovell et al., 2000*) and all-atom torsion angle flexible fitting. Flexible fitting incorporated: (a) full stereochemical restraints using RSRef-CNS; (b) a supplementary flat-bottomed potential to restrain (φ, ψ) backbone dihedrals to the favored areas of a Ramachandran plot; and (c) constrained icosahedral symmetry (*Chapman et al., 2013*; *Brünger et al., 1998*). In early iterations, simulated annealing (5,000K) had some advantage, but later, gradient descent (L-BFGS [*Nocedal, 1980*]) optimization was more effective. Atomic B-factors were refined using a restraint such that the root mean-square deviation between bonded atoms (RMSDB) for the receptor complex was less than in the crystal structure of AAV2 (1.44 $Å^2$) (*Xie et al., 2002*). The first round of modeling and refinement used the unsharpened map which had an effective resolution of $d_{1/2}$ = 3.7 Å, as determined by refining the resolution of a fifth-order Butterworth filter applied to density from the atomic model when fit to the map (*Chapman et al., 2013*). The real-space correlation coefficient (CC) of this intermediate model was 0.84 when using all map grid points within 2 Å of atoms. A second round of modeling and refinement followed sharpening of the reconstruction using the automated procedure in Relion (B = −80.4 $Å^2$). For the start of round 2, the B-factors were reset to 15 $Å^2$, then the same steps were followed, except that solvent and ion atoms were added to AAVR residues 405–499 and AAV 237–735. The final model had a real-space correlation of CC = 0.88 vs. the sharpened map. Following restrained B-factor refinement, RMSDB = 1.24 $Å^2$ with mean AAV2 and AAVR B-factors of 15.2 and 27.7 $Å^2$ respectively. The effective resolution refined to $d_{1/2}$ = 2.3 Å and the model envelope correction to 0.0, in excellent agreement with the $FSC_{0.143}$ = 2.4 Å (*Chapman et al., 2013*).

Biological analysis of the virus-receptor complex was aided by overlaying the contact footprints for PKD2 (on AAV2), epitopes and mutational sites of relevant footprints on projected surfaces of AAV structures. Graphical integration of sequence conservation (*Figure 6d*) was performed using Roadmap (*Chapman, 1993*), while other surface projections (*Figure 6* and *Figure 9*) were calculated using Rivem (*Xiao and Rossmann, 2007*).

## Mass spectrometry

AAV2 capsids were cross-linked to MBP-PKD1-5 with a CID-cleavable, biotinylated cross-linker, cyanurbiotindimercaptopropionylsuccinimide (CBDPS-H8/D8; Creative Molecules, Inc, Cat No 014S). AAV2 and MBP-PKD1-5 (1:54 molar ratio, AAVR:AAV2-subunit) were incubated together for 1 hr at room temperature (RT) to form complexes. CBDPS-H8/D8 cross-linker was added in 84-fold molar excess relative to MBP-PKD1-5 and incubated 30 min at room temperature (RT). (The mixture of AAV2 and MBP-PKD1-5 was in a buffer of 45 mM HEPES, 20 mM $MgCl_2$, 70 mM $NaCl_2$ at pH 7.4. After addition of cross-linker, the buffer at pH 7.4 contained 43 mM HEPES, 19 mM $MgCl_2$, 67 mM $NaCl_2$, 0.04% DMSO, pH 7.4.) Crosslinking was quenched by addition of 500 mM Tris, pH 7.4 (yielding 25 mM Tris) and samples were lyophilized and stored at −20°C. Cross-linked capsid-AAVR complexes were denatured (to increase the subsequent proteolytic digestion) through incubation for 30 min at 70°C in 50 µl 6 M urea, 25 mM Tris, pH 8.0. Dithiothreitol (DTT) was added to 10 mM final concentration and incubated an additional 30 min at 80°C. Iodoacetamide (IAA) was added to a final concentration of 10 mM and incubation continued for 30 more min at RT in the dark. After dilution to 3 M urea, 40 µl 0.1 µg/µl LysC/Tryp (Trypsin/Lys-C Mix, Mass Spec Grade, Promega Cat No V5071) was added and incubated another 4 hr at RT, followed by addition of Glu-C digest buffer (25 mM Tris, pH 7.4) and 40 µl 0.1 µg/µl Glu-C (Glu-C, Sequencing Grade, Promega Cat No V1651) and final incubation at 37°C for 16 hr. The digest reaction was quenched with formic acid.

Peptides from digested AAV2-MBP-PKD1-5 complex were affinity purified to enrich in peptides containing the biotinylated CBDPS cross-linker. Affinity purification was performed using reagents and hardware provided in a Cleavable ICAT Reagent Kit for Protein Labeling (monoplex version) using the

manufacturer's recommended protocol (Sciex). Briefly, peptides were first purified using cation exchange chromatography to remove unbound cross-linker. Then biotinylated peptides were purified using an avidin column, dried by vacuum centrifugation, dissolved in 20 µl of 5% formic acid and analyzed by liquid chromatography/mass spectrometry. Samples were injected onto an Acclaim PepMap 100 µm x 2 cm NanoViper C18, 5 µm trap (Thermo Fisher Scientific), at 5 µl/min for 10 min in mobile phase A containing water, 0.1% formic acid, then switched on-line to a PepMap RSLC C18, 2 µm, 75 µm x 25 cm EasySpray column (Thermo Fisher Scientific). Peptides were then eluted using a 7.5–30% mobile phase B (acetonitrile, 0.1% formic acid) gradient over 90 min at a 300 nl/min flow rate. Data-dependent tandem mass spectrometry analysis was performed using an Orbitrap Fusion instrument fitted with an EasySpray source (Thermo Fisher Scientific). Survey scans (m/z = 400–1500) and MS2 scans (m/z = 100–1800) were performed in the Orbitrap mass analyzer at a resolution = 120,000, and 30,000, respectively, following higher energy collision dissociation (HCD) using a collision energy of 35 following quadrupole isolation at a 1.6 m/z isolation width. Peptides of charge states 3–7 were selected with signal intensities over $5 \times 10^4$ and having a targeted inclusion mass difference of 8.05 to select peptides containing the mass shifted CBDPS cross-linkers. The method also used dynamic exclusion with 30 s duration and mass tolerance of 10 ppm. Cross-linked peptides were identified (*Figure 10*) using StavroX software (version 3.6.0.1) (*Götze et al., 2012*) using cross-linker masses of 509.0974 and 517.1476 for the (H8) and (D8) forms of the CBDPS cross-linker respectively, and mass precision tolerances of 2 and 5 ppm for precursors and fragment ions, respectively.

## Acknowledgements

Sirika Pillay and Jan Carette are thanked for continuing insightful discussions as the project transitioned from receptor identification to structure. EM was performed at: (1) the Multiscale Microscopy Core (MMC) with technical support from the Oregon Health and Science University (OHSU)-FEI Living Lab and the OHSU Center for Spatial Systems Biomedicine (OCSSB); and (2) the Biological Science Imaging Resource at Florida State University. Grant Zane is thanked for help with *Figure 1*. The research was funded, in part by NIH R01 GM066875 (MSC) and R35 GM122564 (MSC).

## Additional information

### Funding

| Funder | Grant reference number | Author |
| --- | --- | --- |
| National Institutes of Health | R01GM066875 | Michael Stewart Chapman |
| National Institutes of Health | R35GM122564 | Michael Stewart Chapman |

The funders had no role in study design, data collection and interpretation, or the decision to submit the work for publication.

### Author contributions

Nancy L Meyer, Guiqing Hu, Omar Davulcu, Data curation, Formal analysis, Validation, Investigation, Visualization, Writing—review and editing; Qing Xie, Craig Yoshioka, Data curation, Formal analysis, Investigation, Writing—review and editing; Alex J Noble, Drew S Gingerich, Investigation; Andrew Trzynka, Resources, Data curation, Software; Larry David, Conceptualization, Resources, Data curation, Formal analysis, Supervision, Validation, Investigation, Visualization, Writing—review and editing; Scott M Stagg, Michael Stewart Chapman, Conceptualization, Resources, Data curation, Software, Formal analysis, Supervision, Validation, Visualization, Methodology, Project administration, Writing—drafting, review and editing

### Author ORCIDs

Nancy L Meyer (iD) https://orcid.org/0000-0002-6836-6688
Alex J Noble (iD) https://orcid.org/0000-0001-8634-2279
Craig Yoshioka (iD) https://orcid.org/0000-0002-0251-7316
Michael Stewart Chapman (iD) https://orcid.org/0000-0001-8525-8585

**Decision letter and Author response**
Decision letter https://doi.org/10.7554/eLife.44707.030
Author response https://doi.org/10.7554/eLife.44707.031

# Additional files

## Supplementary files
• Transparent reporting form
DOI: https://doi.org/10.7554/eLife.44707.016

## Data availability

Electron microscopy maps and atomic coordinates will be available from the electron microscopy and protein data banks (https://www.ebi.ac.uk/pdbe/emdb/ & https://www.rcsb.org/). For the high resolution PKD1-2/AAV2 complex, the accession numbers are EMD-0553, PDB ID 6NZ0, respectively. Reconstructions for the 4 tomographic classes have accession numbers of EMD-0621, EMD-0622, EMD-0623 and EMD-0624.

The following datasets were generated:

| Author(s) | Year | Dataset title | Dataset URL | Database and Identifier |
|---|---|---|---|---|
| Meyer NL, Xie Q, Davulcu O, Yoshioka C, Chapman MS | 2019 | Cryo-EM structure of AAV-2 in complex with AAVR PKD domains 1 and 2 | http://www.rcsb.org/structure/6NZ0 | Protein Data Bank, 6NZ0 |
| Meyer NL, Hu G, Davulcu O, Xie Q, Noble A, Yoshioka C, Gingerich D, Trzynka A, David L, Stagg SM, Chapman MS | 2019 | Cryo-EM structure of AAV-2 in complex with AAVR PKD domains 1 and 2 | https://www.ebi.ac.uk/pdbe/emdb/EMD-0553 | Electron Microscopy Data Bank, EMD-0553 |
| Hu GQ, Meyer NL, Stagg SM, Chapman MS, Davulcu O, Xie Q, Noble AJ, Yoshioka C, Gingerich D, Trzynka A, David L | 2019 | Structure of the AAV2 with its Cell Receptor AAVR | https://www.ebi.ac.uk/pdbe/emdb/EMD-0621 | Electron Microscopy Data Bank, EMD-0621 |
| Hu GQ, Meyer NL, Stagg SM, Chapman MS, Davulcu O, Xie Q, Noble AJ, Yoshioka C, Gingerich D, Trzynka A, David L | 2019 | Structure of the AAV2 with its Cell Receptor AAVR | https://www.ebi.ac.uk/pdbe/emdb/EMD-0622 | Electron Microscopy Data Bank, EMD-0622 |
| Hu GQ, Meyer NL, Stagg SM, Chapman MS, Davulcu O, Xie Q, Noble AJ, Yoshioka C, Gingerich D, Trzynka A, David L | 2019 | Structure of the AAV2 with its Cell Receptor AAVR | https://www.ebi.ac.uk/pdbe/emdb/EMD-0623 | Electron Microscopy Data Bank, EMD-0623 |
| Hu GQ, Meyer NL, Stagg SM, Chapman MS, Davulcu O, Xie Q, Noble AJ, Yoshioka C, Gingerich D, Trzynka A, David L | 2019 | Structure of the AAV2 with its Cell Receptor AAVR | https://www.ebi.ac.uk/pdbe/emdb/EMD-0624 | Electron Microscopy Data Bank, EMD-0624 |

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
