## [Decision Letter]

Thank you for submitting your article "Structure of the gene therapy vector, adeno-associated virus with its cell receptor, AAVR" for consideration by *eLife*.. Your article has now been reviewed by three peer reviewers, and the evaluation has been overseen by Sriram Subramaniam as Reviewing Editor and Cynthia Wolberger as the Senior Editor. The following individuals involved in review of your submission have agreed to reveal their identity: Bernard Heymann (Reviewer #1); Xiao Xiao (Reviewer #2); Peter Tattersall (Reviewer #3).

The reviewers have discussed the reviews with one another and the Reviewing Editor has drafted this decision to help you prepare a revised submission.

All reviewers of the manuscript agree that it represents a major contribution to our understanding of the AAV attachment and entry processes, which are central to its successful use and further exploitation as a gene therapy vector. The reviewers mostly had minor concerns, summarized below. These points need to be addressed before the manuscript can be accepted for publication in *eLife*.

The most significant concern bears on the accessibility of the manuscript to a general audience. The paper is a little hard to read and understand if approached in the order presented. It is difficult to follow the logic of the combination of methodologies and interaction partners used, without first reading the Experimental Design section that starts off the Materials and methods. It would make more sense, to insert this section at the beginning of the Results section and to provide some additional explanation that would be accessible to a general audience. In addition, the implications for the role of antibodies in competing for receptor binding should be explained more explicitly. Readers would also benefit from subheadings that highlight the scientific points in different sections.

The following additional points should be addressed:

1) A simple bar diagram of the constructs, as presented in the Pillay et al., references cited might be helpful. The exact start and finish amino acids, within the full-length AAVR molecule, should be included to allow others to repeat and extend these important studies.

2) Figures 1 and 2 have several small images that could use some labels.

3) In Figure 2, specify which colors correspond to which classes. The classes in panels A-D should also be colored accordingly.

4) Figure 3B is very complicated. Is there a clearer way to show it? In Figure 3D it is impossible to see the color under the hashed footprint.

5) Figure 4 labeling is rather confusing because the black boxes correspondent to the VRI and VRIII regions, but it is not explicitly labeled. This may leave the readers guessing to the identity of the regions shown. Using I and II as labels also may be confusing because readers may believe the boxes correspond to VRI and VRII. Figure 4B and C are also somewhat confusing because the area shown is VRIII but is called region "I" in the legend. Unless region "I" refers to the black box in Figure 4A labeled with "I", the authors may want to figure out a better labeling scheme so it does not confuse the readers.

6) Please show FSC curves and provide masked and unmasked FSC curves for the 2.4 Å map.

7) "Materials and methods are available as supplementary materials at the Science website" – erroneous insertion should be corrected.

8) Please rewrite this sentence in the Discussion to make its meaning clearer and more specific: "Semantics, outdated by recent advances, sow confusion."

9) Figure 1A legend: "AAV" – is this AAV2?

10) "shorter AAVR constructs were used henceforth" – see comment above about the need for diagram.

11) Please explain how "near-saturated binding" was established? What is the stoichiometry at this level of binding? Can the extent of the particle surface involved in adherence to the grid be estimated, and would this be expected to be unavailable for even small AAVR sub-species binding?

12) Results, fifth paragraph: While the high resolution structure may show no evidence of longer range conformational change, one should recognize the caveat that these particles are VLPs, devoid of the genome and N-terminal extensions of the basic structural protein unit. There is significant evidence that these components affect the conformational flexibility of the parvovirus capsid versus virion, i.e. stiffness at the 2-fold, ability to extrude the VP N-termini through the 5-fold channel, etc.

13) Results, sixth paragraph: steric blocking of AAVR binding by neutralizing antibody might not be the only clashes likely at the 3-fold. Although the beautiful AAVR PKD2 footprints in Figure 3 would appear to allow binding at all three positions at the 3-fold, how likely is it that three full-length AAVR ectodomains could be accommodated here?

14) Subsection “Mass spectrometry”, first paragraph: what were the buffer conditions used?

---

## [Author Response]

[…] The most significant concern bears on the accessibility of the manuscript to a general audience. The paper is a little hard to read and understand if approached in the order presented. It is difficult to follow the logic of the combination of methodologies and interaction partners used, without first reading the Experimental Design section that starts off the Materials and methods. It would make more sense, to insert this section at the beginning of the Results section and to provide some additional explanation that would be accessible to a general audience. In addition, the implications for the role of antibodies in competing for receptor binding should be explained more explicitly. Readers would also benefit from subheadings that highlight the scientific points in different sections.

The first three paragraphs of the Materials and methods section, covering overall strategy, have been moved, as suggested, to become the first section of the Results. We thank the reviewers for encouraging the insertion of additional rationalization of the biophysical techniques and receptor constructs. We agree that this will help a general audience appreciate the symbiosis of the multiple approaches pursued.

A paragraph has been inserted prior to the discussion of epitopes, introducing the reasons why overlap of antibody-binding and receptor-binding sites would be of interest. A few sentences have been added at the end of the Discussion to make more explicit the implications both for our understanding of AAV's virology and the implications for gene therapy development.

Sub-headings have been added as requested for subsections of the Results and Discussion sections.

The following additional points should be addressed:1) A simple bar diagram of the constructs, as presented in the Pillay et al., references cited might be helpful. The exact start and finish amino acids, within the full-length AAVR molecule, should be included to allow others to repeat and extend these important studies.

This is inserted as a new Figure 1.

2) Figures 1 and 2 have several small images that could use some labels.

These figures, now Figures 2 and 3 are additionally annotated.

3) In Figure 2, specify which colors correspond to which classes. The classes in panels A-D should also be colored accordingly.

Now as Figure 3, the coloring and captioning is as requested.

4) Figure 3B is very complicated. Is there a clearer way to show it? In Figure 3D it is impossible to see the color under the hashed footprint.

Now as Figure 6, complexity is reduced and clarity improved by the following: Panel B has been separated into panels B through F for each epitope; and hashing has been removed, leaving black outline to designate the AAVR PKD2 footprint.

5) Figure 4 labeling is rather confusing because the black boxes correspondent to the VRI and VRIII regions, but it is not explicitly labeled. This may leave the readers guessing to the identity of the regions shown. Using I and II as labels also may be confusing because readers may believe the boxes correspond to VRI and VRII. Figure 4B and C are also somewhat confusing because the area shown is VRIII but is called region "I" in the legend. Unless region "I" refers to the black box in Figure 4A labeled with "I", the authors may want to figure out a better labeling scheme so it does not confuse the readers.

Now as Figure 8, the boxes have been relabeled (A and B) and the caption now makes clear distinctions between regions in the 3D structure (now called volumes) and parts of the primary sequence that we continue to refer to by the established nomenclature of variable region VR I through VR VIII.

6) Please show FSC curves and provide masked and unmasked FSC curves for the 2.4 Å map.

This has been inserted as a new Figure 5.

7) "Materials and methods are available as supplementary materials at the Science website" – erroneous insertion should be corrected.

The erroneous insertion has been removed.

8) Please rewrite this sentence in the Discussion to make its meaning clearer and more specific: "Semantics, outdated by recent advances, sow confusion."

This sentence has been replaced with: "While our findings contradict some prior conclusions, they are actually consistent with much of the underlying data. Confusion has resulted from the lack of distinction in the historical literature between the attachment of AAV to cells and its entry." Some additional changes have been made in the rest of the paragraph to explain.

9) Figure 1A legend: "AAV" – is this AAV2?

Yes – corrected.

10) "shorter AAVR constructs were used henceforth" – see comment above about the need for diagram.

As previously noted, a domain schematic has been added as a new Figure 1.

11) Please explain how "near-saturated binding" was established? What is the stoichiometry at this level of binding? Can the extent of the particle surface involved in adherence to the grid be estimated, and would this be expected to be unavailable for even small AAVR sub-species binding?

An explanation of the high-saturation binding has been inserted in the first paragraph of the subsection “Single particle cryo-EM for AAV2 complexed with a PKD1-2 construct”. Stoichiometry and grid-based occlusion are discussed later in the second paragraph of the aforementioned subsection, along with implications relevant to point 13 below.

12) Results, fifth paragraph: While the high resolution structure may show no evidence of longer range conformational change, one should recognize the caveat that these particles are VLPs, devoid of the genome and N-terminal extensions of the basic structural protein unit. There is significant evidence that these components affect the conformational flexibility of the parvovirus capsid versus virion, i.e. stiffness at the 2-fold, ability to extrude the VP N-termini through the 5-fold channel, etc.

Absolutely. The suggested caveat has been added to the subsection “Structure at the AAV2-PKD2 binding interface”.

13) Results, sixth paragraph: steric blocking of AAVR binding by neutralizing antibody might not be the only clashes likely at the 3-fold. Although the beautiful AAVR PKD2 footprints in Figure 3 would appear to allow binding at all three positions at the 3-fold, how likely is it that three full-length AAVR ectodomains could be accommodated here?

This comment moves us to be more explicit about the likelihood of additional steric clashes mediated by other domains. Crowding of AAVR around 3-fold axes is further clarified elsewhere with added discussion of the implications of the observed receptor occupancy (point 11 above).

14) Subsection “Mass spectrometry”, first paragraph: what were the buffer conditions used?

The HEPES-based buffers are now described after the second sentence of the Mass Spectrometry section.